# Dynamic Nondestructive Detection Models of Apple Quality in Critical Harvest Period Based on Near-Infrared Spectroscopy and Intelligent Algorithms

**DOI:** 10.3390/foods13111698

**Published:** 2024-05-28

**Authors:** Zhiming Guo, Xuan Chen, Yiyin Zhang, Chanjun Sun, Heera Jayan, Usman Majeed, Nicholas J. Watson, Xiaobo Zou

**Affiliations:** 1China Light Industry Key Laboratory of Food Intelligent Detection & Processing, School of Food and Biological Engineering, Jiangsu University, Zhenjiang 212013, China; cx18756041020@163.com (X.C.); zyysnb123@gmail.com (Y.Z.); chanjun.sun@ujs.edu.cn (C.S.); heerajayan93@outlook.com (H.J.); zou_xiaobo@ujs.edu.cn (X.Z.); 2International Joint Research Laboratory of Intelligent Agriculture and Agri-Products Processing, Jiangsu University, Zhenjiang 212013, China; majeedusman55@gmail.com; 3School of Food Science and Nutrition, University of Leeds, Leeds LS2 9JT, UK; n.j.watson@leeds.ac.uk

**Keywords:** NIR spectroscopy, apple, dynamic detection, machine learning, maturity

## Abstract

Apples are usually bagged during the growing process, which can effectively improve the quality. Establishing an in situ nondestructive testing model for in-tree apples is very important for fruit companies in selecting raw apple materials for valuation. Low-maturity apples and high-maturity apples were acquired separately by a handheld tester for the internal quality assessment of apples developed by our group, and the effects of the two maturity levels on the soluble solids content (SSC) detection of apples were compared. Four feature selection algorithms, like ant colony optimization (ACO), were used to reduce the spectral complexity and improve the apple SSC detection accuracy. The comparison showed that the diffuse reflectance spectra of high-maturity apples better reflected the internal SSC information of the apples. The diffuse reflectance spectra of the high-maturity apples combined with the ACO algorithm achieved the best results for SSC prediction, with a prediction correlation coefficient (Rp) of 0.88, a root mean square error of prediction (RMSEP) of 0.5678 °Brix, and a residual prediction deviation (RPD) value of 2.466. Additionally, the fruit maturity was predicted using PLS-LDA based on color data, achieveing accuracies of 99.03% and 99.35% for low- and high-maturity fruits, respectively. These results suggest that in-tree apple in situ detection has great potential to enable improved robustness and accuracy in modeling apple quality.

## 1. Introduction

In recent years, apple, as a highly favored fruit, has shown continuous growth in both global production and consumption [1]. Apples are bagged during growth to prevent and remove fruit rust and surface blemishes, promote coloration, reduce pesticide residues, and improve storability [2]. Apples are removed from their protective bags about 14–20 days before the harvest period, which is very important for surface coloring and sugar conversion. It is also important for fruit harvesting companies to select apple materials for valorization and harvesting. However, current studies are still limited to the static quality inspection of fruits, while in-depth studies on the dynamic changes in apple quality during the harvesting period are relatively limited [3]. In fact, apples undergo a series of physiological and chemical changes after harvest, such as respiration and sugar metabolism, processes that directly affect their taste and storage life [4]. Therefore, establishing a dynamic nondestructive testing model for apple quality during harvesting is of great practical significance in realizing the comprehensive monitoring of fruit production, storage, and marketing.

A commonly used technique to achieve this goal is near-infrared (NIR) spectroscopy, because of its efficient and nondestructive nature [5,6,7]. With the development of chemometric methods and the availability of instruments, NIR spectroscopy has been widely used for the determination of fruit SSC [8]. Research in the field of fruit detection using NIR spectroscopy mainly focuses on establishing spectral signal models and developing corresponding processing methods. In terms of model building, intelligent algorithms such as principal component analysis (PCA), partial least squares (PLS), and support vector machines (SVM) have been realized to construct static models for the nondestructive testing of fruit quality [9,10,11,12]. Numerous researchers have conducted studies indicating that the fruit quality may change significantly due to biological variability, which affects light propagation and light–substance interactions. The predictive power of the modeling may be reduced as a result [13]. Peirs et al. [14] found that the orchards, seasons, and varieties have a great influence on the spectral variation of apples.

The near-infrared spectra of reflectance modes are more readily available and have relatively high intensity levels [15]. Several studies have been conducted to measure SSC online using reflectance modes, such as those focusing on blueberries [16], oranges [17], and apples [18], suggesting that the reflectance mode is suitable for the accurate determination of fruit’s internal qualities. Therefore, in this study, the diffuse reflectance mode was used to detect the diffuse reflectance spectra of apples online. Fruit ripening and senescence are complex physiological as well as biochemical processes. The physicochemical compositions of fruits vary greatly with the maturity, which may affect the robustness of SSC and hardness calibration models [19]. The fruit maturity indicators are the soluble solids, titratable acid, and color. Shao et al. [20] investigated the relationship between soluble solids and the near-infrared hyperspectral data of medium-ripe and fully ripened winter jujubes. Yu et al. [21] constructed a quantitative maturity model for Kurrer balsam pears, choosing the hardness, soluble solids content, and titratable acid to quantify the maturity of the pears. Pourdarbani et al. [22] classified “Fuji” apples into four stages of maturity based on color and spectral data. Fruits at the freshly bagged and uncolored stage exhibit lower maturity, and those harvested when they reach the harvest standard demontrate higher maturity. Zhang et al. [23] collected spectral data on apples to evaluate their maturity levels. On the other hand, DeLong et al. [24] used a handheld chlorophyll meter to obtain chlorophyll content data on apples, and the developed model was used to succesfully assess the harvest maturity of “Minneiska” apples. Therefore, the nondestructive testing of dynamic change processes in new application scenarios is necessary to investigate the effect of ripeness on apple quality detection during the harvesting period. The goal is to establish a quality detection model that is insensitive to changes in apple ripeness while on the tree.

The purpose of this study is to model the dynamic nondestructive detection of apples on trees during the harvesting period after the removal of apple bags. The technical approach is depicted in Figure 1. The specific objectives are to (1) compare and analyze the change rules of the apple surface coloring and brix on the tree during the harvesting period and establish a mechanism linking near-infrared spectroscopy and different maturity levels; (2) analyze and compare the optimization results of reflectance spectral feature algorithms for two maturity levels of apples before and after the harvesting period, to provide a recommended solution for the dynamic quality inspection of apples; and (3) study the effect of color parameters (L*, a*, b*) on the modeling of the dynamic change process. A classification model was built with PLS-LDA.

## 2. Materials and Methods

### 2.1. Sample Collection

The apples used in this experiment were produced in an orchard at an 80°16′ eastern longitude and 41°18′ northern latitude in the Aksu region, Xinjiang. The Fuji variety apples were similar to those available in the market and represented the predominant cultivar in the area. The color of the sun-exposed and shaded sides of the apples was determined when the apples were first bagged, and the spectra of the apples were detected in situ on the tree using a prototype handheld monitoring terminal developed by our group. The apples were then collected to determine their physicochemical information. Then, 14 days later, when the harvest standard was reached, as identified by a professional harvester, samples that were relatively the same size as those of the low-maturity apples and had no scars on their surfaces were detected in situ on the tree. The harvest dates and numbers of samples at different maturity stages are shown in Table 1. The sample surfaces were wiped with damp gauze and allowed to dry naturally. Then, the samples were numbered one by one and the spectral and physicochemical information of the samples was collected.

### 2.2. Color Measurement

The color of the samples was measured with a spectrophotometer (CM-2300d, Konica Minolta, Osaka, Japan). In the CIE L*a*b* color space, the color is estimated by three parameters: L* denotes the luminance from black (0) to white (100); a* denotes the green–red trend, with negative values toward green (−) and positive values toward red (+); and b* denotes the blue–yellow trend, with negative values toward blue (−) and positive values toward yellow (+) [25,26,27]. The spectrophotometer was first calibrated with a standard white calibration plate CM-A145. Then, two separate equatorial measurements were taken of the sunward and dorsal surfaces of the sample. The color parameters of the sunward and dorsal sides of the samples were averaged over the two measurements.

### 2.3. Measurement of Apple SSC

The SSC measurement procedure was as follows. After removing the peel, a 10 × 10 × 10 mm tissue block was removed at the equator of the apple with a sampler. The SSC of the apples was determined by squeezing the juice from the removed tissue blocks wrapped in gauze. A portable refractometer (PAL-1, Atago, Tokyo, Japan) with automatic temperature compensation was used. The determination of the suspended solids in the apples was carried out according to the agriculture industry standard (NY/T2637-2014). The refractometer was first zeroed with pure water and then the squeezed juice was dropped onto the test lens. The refractometer prism was cleaned with pure water at the end of each test. The average of the two measurements was considered as the SSC value for each sample.

### 2.4. Prototype of Handheld Detection Terminal

The internal quality of a sample can be detected using the apple internal quality handheld tester. As shown in Figure 2, the apple internal quality handheld tester and an Internet of Things (IoT) cloud data system comprise the detector. When testing, the apple internal quality handheld tester can acquire the reflectance spectrum information of the apple. At the same time, the data are transferred to the IoT cloud data system via a communication module. In the Internet of Things (IoT) cloud data system, the model of the inspection is used for calculation and can predict the inspection results. Data can also be queried, downloaded, and statistically analyzed in the detection database.

The hardware of the handheld inspection terminal of the test system mainly consists of a light-emitting diode (LED) light source, a near-infrared photoelectric sensor, a light-shielding ring, a rubber gasket, a temperature sensor, a casing, a rechargeable lithium battery, a control circuit, a display, and so on. The LED point light source is used to irradiate the tested apple. The rubber gasket protects the apple from mechanical damage. In addition, the shading ring excludes stray light other than diffuse reflected light, fully ensuring that it is not affected by ambient light during outdoor inspection. During the detection process, the light illuminates the apple and then passes through a series of internal transmissions and is finally received by the visible near-infrared photoelectric sensor. After the control circuit processes the received signal, it is then transmitted to the cloud server through the 4G/5G module. The final detection results can be calculated by using the cloud model.

In addition, the apple internal quality handheld detector measures 118 × 50 × 38 mm and weighs about 0.12 kg. It has the advantages of a small size, high integration, low energy consumption, and ease of secondary development, making it easy to carry and more convenient for apple quality assessment. Compared with traditional portable and handheld NIR detection devices in the market, it has a greatly simplified design structure and significantly reduced cost. In conclusion, the advantages of this design are low power consumption, a low cost, and ultra-portability. It can be used by fruit companies, farmers, and apple producers for a wide range of applications.

### 2.5. NIR Spectra Acquisition

The diffuse reflectance spectra of the samples were collected using the apple internal quality handheld detector, as described in the previous section. After starting the spectral acquisition software, as optimization was performed many times before this experiment, the average number of times was set to 5, the exposure time was 75 ms, the sampling frequency was 10 Hz, and the pulse-width modulation (PWM) value was 60. The apple was attached via the equatorial position to the rubber washer, and, after pressing the detection button, the miniature light source began to flash and the spectral data appeared in the spectral acquisition software. The spectral acquisition was completed upon clicking the save button. Two measurements were taken at the equator of each sample on the sun-exposed side and the shaded side. Our operations were conducted on the surfaces of the apples, eliminating the need for peeling. In order to resolve the testing errors caused by dark noise, a total reflection reference spectrum was acquired using a PTFE reflector before the apple spectrum acquisition. In addition, dark noise spectra were acquired simultaneously with the apple spectra and the diffuse reflectance spectral intensity was converted into the relative absorbance lg(1/R).
$$R=\frac{I_{\lambda }-D_{\lambda }}{R_{\lambda }-D_{\lambda }}$$
where R is the corrected spectrum; I_λ_ is the original apple spectrum collected; D_λ_ is the dark noise spectrum; and R_λ_ is the total reflection reference spectrum.

### 2.6. Data Processing Algorithms

#### 2.6.1. Spectral Preprocessing

In the process of acquiring apples’ diffuse reflectance spectra, the spectral detection performance may be affected by the influence of instrument errors, manual manipulation, and variations in the apple samples. To eliminate these effects, spectral preprocessing was performed. Savitzky–Golay smoothing (SG) was first applied for noise reduction, and then attempts were made to further process the apple spectra using the standard normal variable transform (SNV), multiple scattering correction (MSC), normalization (center), first-order differential (1^st^D), and further processing [28].

#### 2.6.2. PLS Model

The partial least squares method is the most classical of the chemometric methods for visible/near-infrared spectral analysis. The partial least squares algorithm first decomposed the apple spectral matrix and the SSC matrix and obtained the score matrix and the loading matrix from these two original matrices [29,30]. In turn, a one-way linear regression relationship between the score matrix of the spectral variable and the score matrix of the SSC variable was established.

#### 2.6.3. Feature Variable Selection

The composition of food is usually complex, with a large number of second and third overtones in the visible/near-infrared region. Interactions between overtone bands result in spectra containing a large amount of irrelevant information. These factors affect the accuracy and robustness of spectral prediction models, and the speed of the detection model may be reduced as a result [31,32]. The four feature variable selection algorithms used in this experiment were the synergetic interval (SI), the continuous projection algorithm (UVE), competitive adaptive reweighted sampling (CARS), and the ant colony optimization algorithm (ACO). The SI algorithm divides the entire spectrum into different subintervals and then combines these different subintervals to find a combination that gives the model the highest accuracy [33,34,35]. To simplify the model, the interval with the smallest RMSEC is chosen during construction. UVE excludes variables considered “useless” from the dataset through thresholds such as variance and the correlation coefficient. By excluding irrelevant variables, redundant and irrelevant information in the data can be effectively reduced, thus helping to improve the performance of the model. CARS evaluates and ranks the importance of each feature and obtains a subset of optimal feature variables. Compared with some other traditional feature selection methods, the CARS algorithm handles the interrelationships between features better, improving the accuracy and stability of feature selection [36]. ACO-PLS is a powerful variable selection algorithm used to solve regression problems. It mimics the behavior of ants in mapping food channels, capturing the target portion of the spectrum from the spectral data. Firstly, it initializes the pheromone vector and other related parameters. Then, it initiates the ants and uses the Monte Carlo roulette wheel spinning method to randomly select a variable from the set until the number of variables reaches the maximum. After completing the variable selection in stages, it establishes a partial least squares model and outputs the root mean square error. All four algorithms are widely used in fruit spectral feature selection.

#### 2.6.4. Model Performance Evaluation

Feature variable selection and PLS regression were performed using MATLAB R2017b (MathWorks, Natick, MA, USA). Then, the parameters of the PLS model were imported into the software developed by our group and five preprocessing methods were used to predict the SSC of the samples with different maturity levels. The correlation coefficient of calibration (Rc), root mean square error of calibration (RMSEC), correlation coefficient of prediction (Rp), root mean square error of prediction (RMSEP), and residual prediction deviation (RPD) were used to compare and evaluate the performance of the model [37]. The degree of linear correlation between the predicted SSC values and true SSC values is indicated by Rc and Rp. RMSEC and RMSEP indicate the deviation between the predicted SSC and true SSC. Higher Rc and Rp values and lower RMSEC and RMSEP values indicate the excellent performance of the established PLS model. The RPD value is the standard deviation divided by the RMSEP. An RPD value >1.8 indicates good model stability and an RPD value > 2 indicates that the predictive accuracy of the model is excellent.

#### 2.6.5. PLS-LDA Classification Model

Partial least squares–linear discrimination analysis (PLS-LDA) is a multivariate analysis technique mainly used to deal with classification and discrimination problems; it combines the ideas of partial least squares regression (PLS) and linear discriminant analysis (LDA) [38]. PLS-LDA is a supervised discriminant analysis method where the dataset contains both an independent variable (X) and a dependent variable (Y), with the ability to model high-dimensional and high-correlation samples that capture the maximum variance of X associated with Y in the space.

## 3. Results

### 3.1. Apple Quality at Different Maturity Stages

#### 3.1.1. Comparison of Color Parameters

The pigments in the skin of red apples consist mainly of chlorophyll, carotenoids, and anthocyanins. Among them, chlorophyll contributes to the green color, carotenoids mainly exhibit a yellow color, and anthocyanins are responsible for the red color of the fruit. The three major classes of pigments exhibit their own dynamic formation patterns during fruit development. Chlorophyll and carotenoids mainly form during the early stage of fruit growth and development, while the formation of pericarp anthocyanins initiates gradually as the fruit ripens, often serving as a marker for the onset of fruit ripening. As shown in Table 2, as the harvest period progresses, the maturity of apples gradually increases. Correspondingly, the L* value gradually decreases, the a* value increases, and the b* value gradually decreases, culminating in the gradual formation of a red color in apples. The a*/b* ratio and the a* value are directly correlated to the anthocyanin content.

#### 3.1.2. SSC Measurement Statistics

The SSC ranges of the low- and high-maturity apples were 9.3–17.0 °Brix and 11.7–19.4 °Brix, respectively. The mean SSC of the low- and high-maturity apples was 13.4 ± 1.2 °Brix and 15.1 ± 1.4 °Brix, respectively. After removing the outliers using Monte Carlo outlier detection, accurate predictions were obtained, as shown in Table 3. The apple samples were randomly divided into calibration and prediction datasets, with a ratio of 3:2. Calibration and prediction sets are independent of each other and do not overlap. On the other hand, outliers in normal samples can have a significant impact on the intelligent algorithms building the model, leading to a decrease in the accuracy of the model predictions.

### 3.2. Spectral Analysis and Pretreatment

The raw spectra were converted into reflectance spectra based on white and dark references. Figure 3 shows the reflectance spectra of the low- and high-maturity apples, respectively. The principal component analysis of the raw spectra of the low- and high-maturity apples is shown in Figure 3c. As can be seen from the figure, the calculated scores of the first three principal components account for 99.57% of the spectral information. There is less overlap between the spectra of the low- and high-maturity apples, indicating that the difference in the reflectance spectra of the samples collected at the two maturity levels is more significant. The effective spectral range was chosen to be 608–958 nm to avoid noise at low energy. The wider absorption range showed significant variation in the peel pigmentation content among apples at different maturity. Both the low- and high-maturity apples’ reflectance spectra had a chlorophyll-related peak at 675 nm. The trough at 710 nm may contain important information related to fruit maturity [39], which may affect the SSC values of the apples. The absorption peak near 750 nm may be the overtone of the O-H band in water. The absorption peak at 820 nm is associated with the N-H overtone [40].

As shown in Table 4, the best performance was seen for the reflectance spectra of highly mature apples combined with the SG-SNV preprocessing model, with Rp, RMSEP, and RPD values of 0.8350, 0.7876 °Brix, and 1.778. Therefore, SG-SNV was chosen as the preprocessing method before feature selection for the reflectance spectra. In the low-maturity apples’ reflectance spectra, the SG-SNV method did not have as high an RPD value as SG-1^st^D, but it was also chosen for preprocessing in order to control the variables to obtain a good model of the reflectance spectra.

### 3.3. Spectral Models Based on Feature Selection

#### 3.3.1. SI-PLS Model Results

The entire spectrum was decomposed into 10 intervals and combined with four subintervals to build the PLS model with accurate prediction. The best prediction results were obtained when the combination had the lowest interval RMSEP. For low-maturity apples, the best combination of selected subintervals was [2 4 7 10], corresponding to 644–678, 716–750, 824–856, and 926–958 nm. For high-maturity apples, the best spectral subintervals were [5 8 9 10], corresponding to 752–786, 858–890, 892–924, and 926–956 nm. The SI-PLS model predictions are shown in Figure 4b and Figure 5b. The RPD values of the low- and high-maturity apples’ spectral models were 1.729 and 1.733. The spectral model of the low-maturity apples showed less accuracy compared to that of the high-maturity apples.

#### 3.3.2. UVE-PLS Model Results

During UVE, the spectral matrix can be cross-validated to eliminate invalid wavelength variables. UVE measures the characteristic wavelength band via controlling the coefficient of variation. Fourteen and twenty-eight wavelength variables were selected in the reflectance spectra of low- and high-maturity apples, respectively. The selected wavelength variables are shown in Figure 4c and Figure 5c. The Rp and RMSEP of the low-maturity apples’ reflectance spectral model were 0.8420 and 0.6810 °Brix, respectively. The RPD values of the low-maturity apples’ and high-maturity apples’ reflectance spectral models were 1.762 and 1.819, respectively. An improvement in the accuracy of the reflectance spectral models of the apples for both maturity levels was realized compared to the full-spectrum model.

#### 3.3.3. CARS-PLS Model Results

Fifty Monte Carlo samplings were set up and the optimal model was determined using five-fold cross-validation based on the RMSEC minimum. The number of samples was gradually increased, the number of selected variables was gradually decreased, and the speed of both the coarse and fine selection processes varied from fast to slow. When the sampling was performed 26 times and 4 times, respectively, the RMSEC value was the smallest, and then it increased gradually. The established models for Rc, RMSEC, Rp, and RMSEP are shown in Figure 4f and Figure 5f. The RPD values of the reflectance spectral models for low- and high-maturity apples were 1.725 and 2.071. However, the RPD value of the reflectance spectral model for high-maturity apples was >2.

#### 3.3.4. ACO-PLS Model Results

The characteristic wavelength points were extracted from the full spectrum using the ACO algorithm. The control parameters of the ACO optimization algorithm were verified by several experiments, as follows: the initial population size was 100, the maximum number of iterations was 50, the maximum number of loops was 10, the probability threshold of variable selection was 0.3, and the significance factor Q was 0.01. The pheromone attenuation coefficient p was 0.65, which reliably ensured the comprehensiveness of the transmitted information and the convergence speed of the algorithm. The selected wavelength variables are shown in Figure 4g and Figure 5g, and the established models for Rc, RMSEC, Rp, and RMSEP are shown in Figure 4h and Figure 5h. The RPD values of the reflectance spectral models for low- and high-maturity apples were 1.868 and 2.466. ACO-PLS achieved the best results in the reflectance spectral modeling of apples at both maturity levels.

### 3.4. Comparisons of Different PLS Models

Figure 4 and Figure 5 show the modeling results of the low- and high-maturity apples’ reflectance spectra processed through the SI, UVE, CARS, and ACO algorithms. In the collection of the low-maturity apples’ reflectance spectra, the Rp values were all less than 0.85 and the RPD values were all less than 2. Meanwhile, in the high-maturity apples’ reflectance spectra, the RP values were greater than 0.80; the RPD values were greater than 1.8 after processing with the UVE, CARS, and ACO algorithms; and the RPD values were greater than 2 after processing with the CARS and ACO algorithms. These suggest that high-maturity apples can be predicted more accurately with respect to the internal SSC values. The performance of the reflectance model for low-maturity apples, in descending order, was ACO-PLS > UVE-PLS > SI-PLS > CARS-PLS > PLS. The performance of the reflectance model for high-maturity apples, in descending order, was ACO-PLS > CARS-PLS > UVE-PLS > SI-PLS > PLS. The ACO-PLS model with reflectance spectra had the best prediction ability for high-maturity apples’ SSC content. Meanwhile, the reflectance spectral model for high-maturity apples had better performance overall.

### 3.5. PLS-LDA Classification Model

To investigate the effect of color on maturity, the color space coordinates (L*, a*, b*) were used as input variables, and then a PLS-LDA classification model was built by combining these variables with the SSC. The SSC array was set to −1 for samples with low maturity and 1 for samples with high maturity. The classification accuracy of the training set is presented in Figure 6. The accuracy of the training and prediction sets was 99.03% and 99.35%, respectively. Overall, the PLS-LDA model was effective in analyzing the apple color during ripening.

The diffuse reflectance spectra corresponding to the classified color coordinates (L*, a*, b*) were preprocessed with SG-SNV and then subjected to PLS modeling. The classified low-maturity apples had more acccurate values, with Rp = 0.8139, RMSEP = 0.7013, and RPD = 1.711, compared to the PLS model. After classification, high-maturity apples, with Rp = 0.8379, RMSEP = 0.7760, and RPD = 1.804, showed slight improvements in the model predictions. The results obtained via the LDA method showed reliability, and it could be applied to different classification methods supervised by discriminant analysis.

## 4. Discussion

The SSC was obtained for two maturity levels of apples by in situ testing on trees. Diffuse reflectance spectra were collected with a handheld inspection system developed by our group for the internal quality assessment of apples, aiming to model the dynamic in situ inspection of apples on trees for the period from bag removal to harvest. Due to the differences in the light transmission properties and cellular structures of apples, the original spectral trends of the apples at different maturity levels were generally similar, but the spectral intensities varied greatly from 650 nm to 800 nm.

Comparing the prediction results of the spectra of the two maturity groups of apples, the SSC prediction accuracy for high-maturity apples was higher than that for low-maturity apples among the various feature extraction methods. There are many reasons for the differences between the modeling results of low- and high-maturity apples. The signals collected by the spectrometer in reflectance mode arise from the apple surface, and prediction models developed by applying NIR spectroscopy for the non-invasive detection of the fruit SSC are often influenced by biological variability factors. Fruits with different physiological stages, varieties, growing regions, or seasons undergo changes in physical properties and chemical composition, which lead to differences in the fruit’s cellular structure and optical properties.

The apple samples were randomly divided into mutually independent calibration and prediction sets. Among the five pretreatment methods (SG, SNV, MSC, center, and 1^st^D), SG and SNV performed better. This was due to the fact that these two methods can eliminate the effects of the solid particle size, surface scattering, and light path variations from the spectra. Among the four feature wavelength extraction methods, the SI algorithm performed the worst, which may have been due to the fact that the optimal subintervals selected were not within the effective spectral range and did not effectively represent the SSC features. Compared to the CARS algorithm, the UVE algorithm removes SSC-independent variables from the spectral information. Most of the characteristic wavelength variables selected by the CARS algorithm were related to the internal chemical properties of the apples, seeking to improve the performance of the model, but potentially valid information was also removed. Compared to the other three spectral feature selection algorithms, the ACO algorithm achieved the best optimization results, and the highest accuracy for SSC prediction model was achieved using the high-maturity apple reflectance spectral model built by the ACO algorithm. This may have been due to the ability of the ACO algorithm to screen out highly competitive combinations of feature wavelengths. The ACO algorithm adaptively adjusted the search behavior through pheromone updating and showed strong adaptability to environmental changes, enabling it to consistently find a better solution. Meanwhile, the best results were achieved by using diffuse reflectance spectroscopy combined with the ACO algorithm to predict the SSC of apples at high maturity, with Rp, RMSEP, and RPD values of 0.88, 0.5678 °Brix, and 2.466, respectively. The PLS-LDA model was more effective regarding the apple color against ripeness, with accuracy of 99.03% for the training set model and 99.35% for the prediction set model. The post-spectral color classification showed great potential in predicting the apples’ SSC.

In addition, the model has the potential to assess the external color and internal quality for the grading of apples for customer satisfaction. These tested and graded apples can better satisfy customers’ purchasing needs and improve the efficiency and accuracy of apple valuation and harvesting.

## 5. Conclusions

This study acquired the diffuse reflectance spectra of low-maturity and high-maturity apple samples using a handheld inspection system for the internal quality assessment of apples developed by our group. Four feature wavelength selection algorithms, namely SI, UVE, CARS, and ACO, were employed for feature wavelength selection. The established ACO-PLS models for high-maturity apples achieved the best results in this study. The model’s Rp, RMSEP, and RPD values were 0.88, 0.5678 °Brix, and 2.466, respectively. In order to consider the influence of color factors on the predictive ability of the model for low- versus high-ripeness apples during the apple harvest period, a PLS-LDA classification model for apple ripeness based on color was developed. The accuracy of the training set model was 99.03% and the accuracy of the prediction set model was 99.35%. The results showed that the new application scenario of dynamic inspection has great potential to achieve improved robustness and accuracy in modeling apple quality during the harvest period. This method can be expanded to other varieties of apples in the future to verify the generalization and applicability of the model.

## Figures and Tables

**Figure 1 foods-13-01698-f001:**
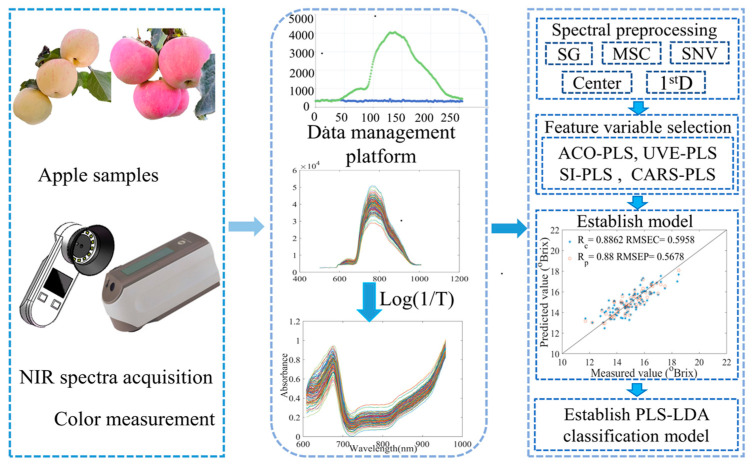
Detecting apple SSC and constructing a dynamic quality detection model for the apple harvesting period using a handheld tester.

**Figure 2 foods-13-01698-f002:**
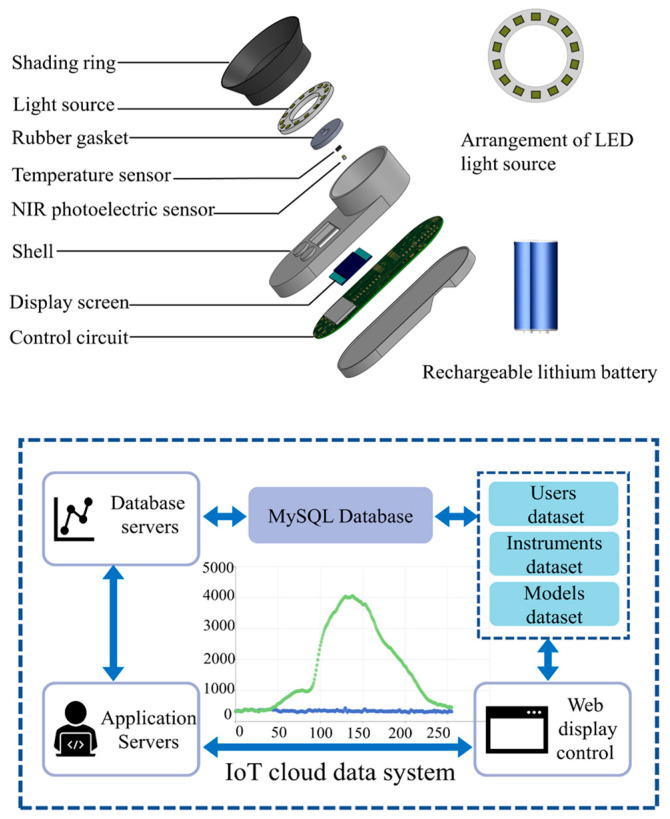
Structure of apple internal quality handheld tester and IoT cloud data testing system.

**Figure 3 foods-13-01698-f003:**
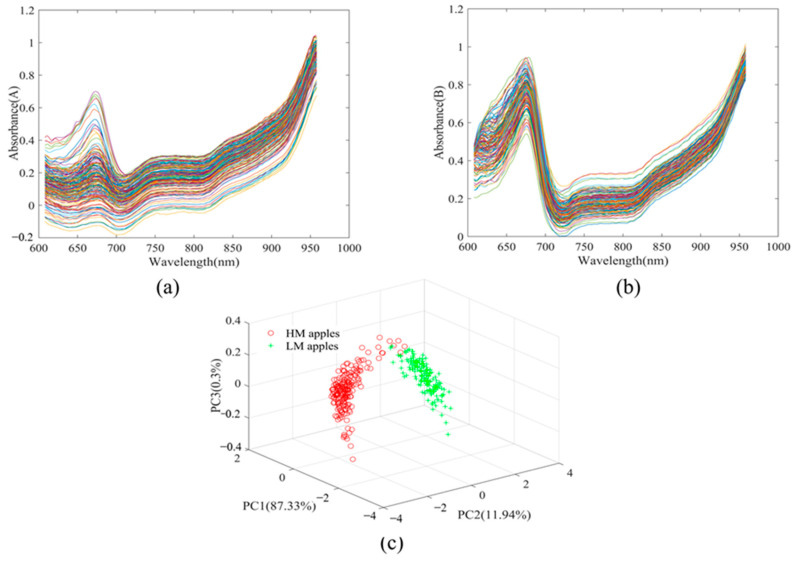
Spectral analysis of apple samples. (**a**) Spectrum of LM apples, (**b**) spectrum of HM apples, (**c**) PCA scatter plot of LM apples (green +) and HM apples (red ○).

**Figure 4 foods-13-01698-f004:**
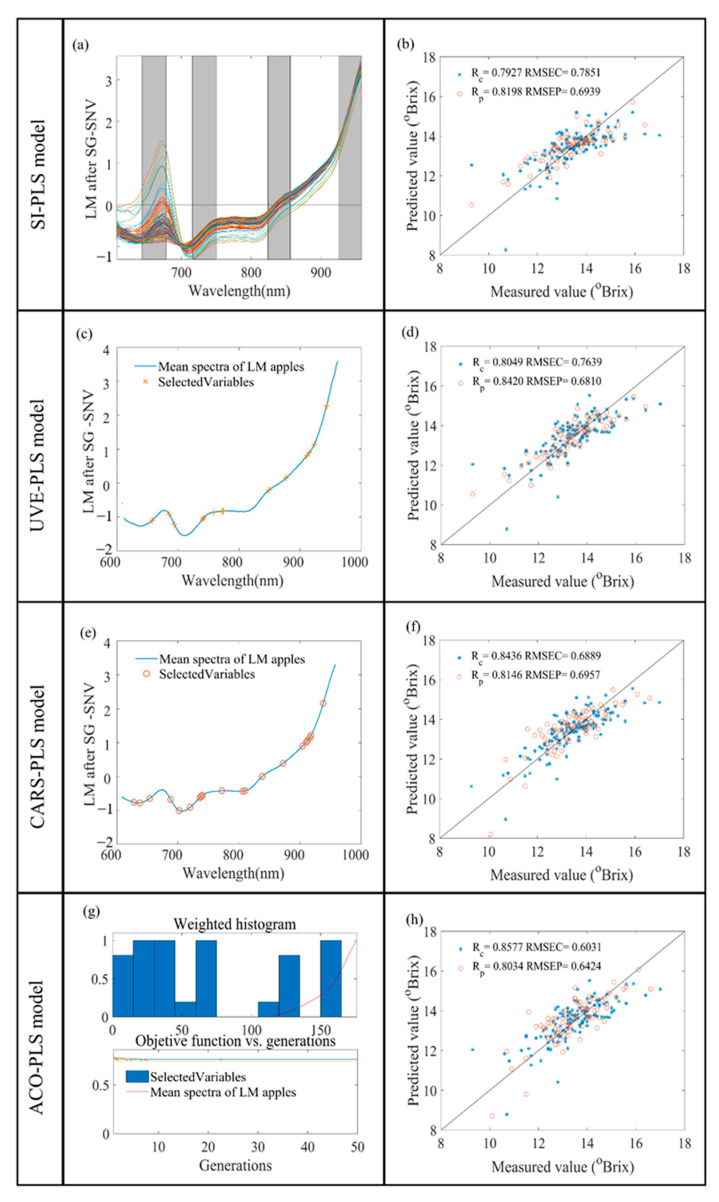
Results of SI, UVE, CARS, and ACO variable selection for LM apples (**a**,**c**,**e**,**g**). The PLS model built by the four variable selection algorithms demonstrates the relevance of the model (**b**,**d**,**f**,**h**) using the distributions of the calibration set (*) and the prediction set (○).

**Figure 5 foods-13-01698-f005:**
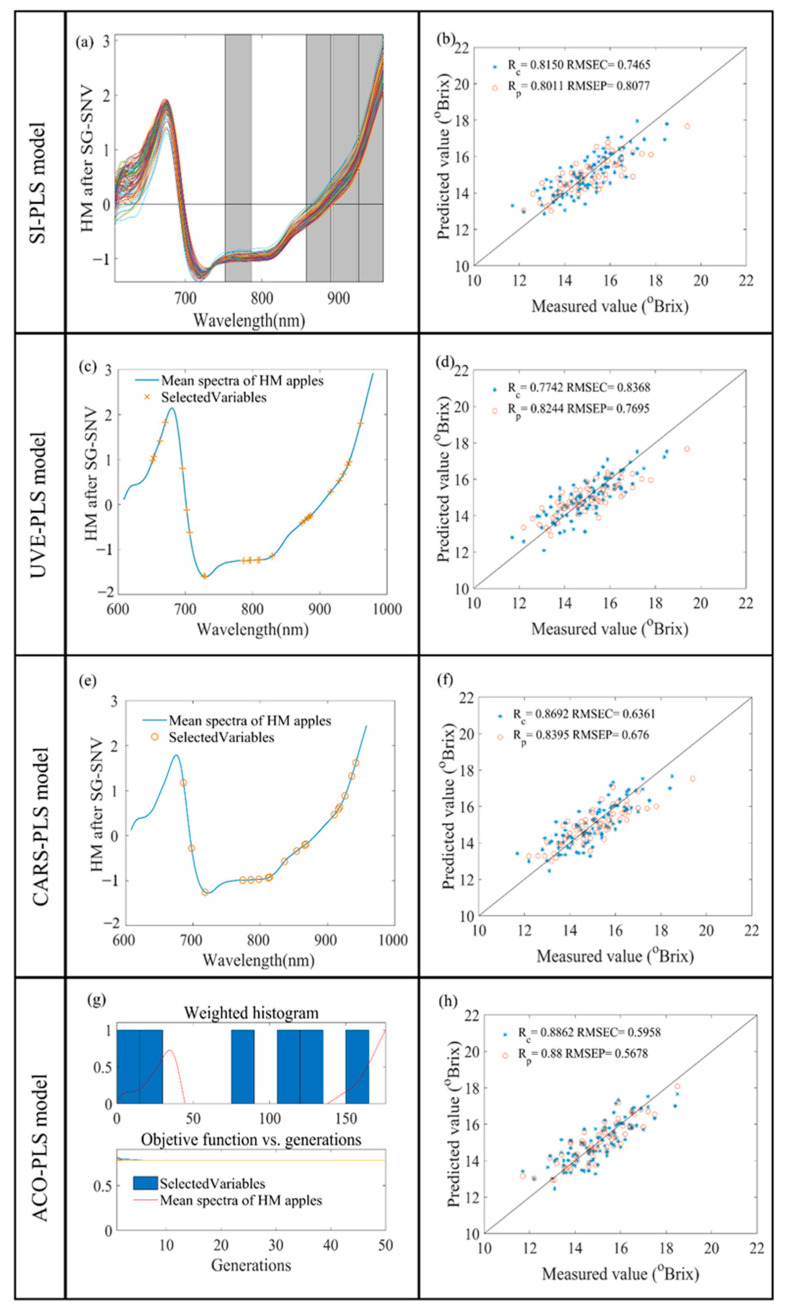
Results of SI, UVE, CARS, and ACO variable selection for HM apples (**a**,**c**,**e**,**g**). The PLS model built by the four variable selection algorithms demonstrates the relevance of the model (**b**,**d**,**f**,**h**) using the distributions of the calibration set (*) and the prediction set (○).

**Figure 6 foods-13-01698-f006:**
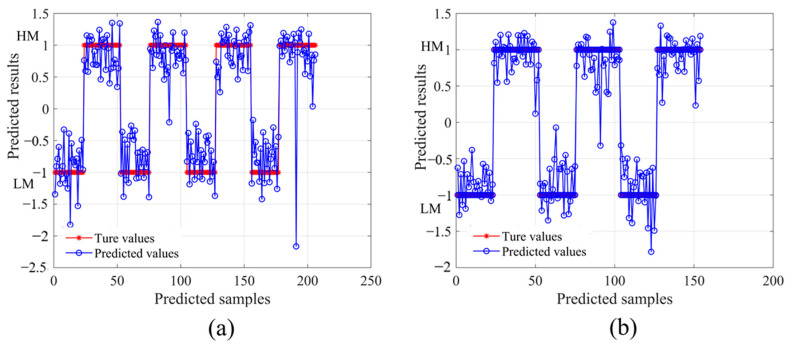
PLS-LDA modeling results with color space coordinates (L*, a*, b*) as input variables, true values (*), and predicted values (○). Comparison of prediction results of training set (**a**) with 99.03% accuracy and comparison of prediction results of prediction set (**b**) with 99.53% accuracy.

**Table 1 foods-13-01698-t001:** Harvest date and number of samples at different maturity stages.

Date	Samples	Maturity
4 October 2023	100	LM
5 October 2023	100	LM
28 October 2023	80	HM
29 October 2023	80	HM

**Table 2 foods-13-01698-t002:** Measured color change statistics of apple samples.

Parameter	Maturity	Samples	Range	Mean
L*	LM	200	39.6–84.38	78.45 ± 4.29
	HM	160	26.41–82.26	68.69 ± 8.00
a*	LM	200	−5.16–27.87	2.46 ± 5.77
	HM	160	−2.41–35.51	13.84 ± 8.89
b*	LM	200	16.31–88.90	27.77 ± 6.16
	HM	160	15.15–40.60	26.60 ± 4.87

**Table 3 foods-13-01698-t003:** Statistics of apple SSC values measured for spectral acquisition methods.

Maturity	Dataset	Samples	Range (°Brix)	Mean (°Brix)
LM	Total	200	9.3–17.0	13.4 ± 1.2
	Calibration	120	9.3–17.2	13.2 ± 1.3
	Prediction	80	9.6–16.8	13.2 ± 1.0
HM	Total	160	11.7–19.4	15.1 ± 1.4
	Calibration	96	11.7–18.5	14.8 ± 1.3
	Prediction	64	12.2–19.4	15.0 ± 1.4

**Table 4 foods-13-01698-t004:** SSC prediction results of PLS LM apples and HM apples with different spectral preprocessing methods.

Maturity	Preprocessing	Samples	Rc	RMSEC	Rp	RMSEP	RPD
LM	SG	200	0.7703	0.7551	0.7916	0.7493	1.601
	SG-SNV		0.7649	0.7639	0.7919	0.7414	1.619
	SG-MSC		0.7753	0.7462	0.7743	0.7676	1.563
	SG-Center		0.7703	0.7551	0.7916	0.7493	1.601
	SG-1^st^D		0.7957	0.7163	0.7977	0.7219	1.662
HM	SG	160	0.7988	0.7954	0.7801	0.8220	1.703
	SG-SNV		0.8356	0.7932	0.8350	0.7876	1.778
	SG-MSC		0.8343	0.7961	0.8294	0.8022	1.745
	SG-Center		0.8456	0.7709	0.8206	0.8186	1.710
	SG-1^st^D		0.8439	0.7747	0.8029	0.8538	1.640

## Data Availability

The original contributions presented in the study are included in the article, further inquiries can be directed to the corresponding author.

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
