# Peer review of "Dynamic Nondestructive Detection Models of Apple Quality in Critical Harvest Period Based on Near-Infrared Spectroscopy and Intelligent Algorithms"

_foods, 2024, doi:10.3390/foods13111698_

Round 1

Reviewer 1 Report

Comments and Suggestions for Authors

The article entitled "Dynamic nondestructive detection models of apple quality in critical harvest period based on near infrared spectroscopy and intelligent algorithms " presents a very interesting study and proposes a green and portable method to determine the quality of apples in situ. However, the description of the different sections of the article is not clear. The authors write in a very confusing way, making it very difficult to read and understand many of the sections, starting with the abstract and ending with the discussion. The authors should write it more clearly so that it can be fully understood.

Nevertheless, here are some of my doubts and specific questions.

1.- In line 20 the acronym "SSC" appears for the first time and there is no indication of what it means. I assume it means "Soluble Solids Content".  It is an important parameter in the development of the article and at no time do the authors indicate what it means.

2.- Line 28 indicates that the colour is measured by PLS-LDA. As far as I know, PLS-LDA is not an analytical technique for measuring colour.

3.- Lines 95-97: In the study, are samples from localized production and samples purchased in supermarkets used?

4.- Lines 115-116: Indicate characteristics of the black and white standard. Indicate why it is necessary to calibrate? I think the authors are referring to adjusting the equipment.

5.- Section 2.5: It is not clear if it is necessary to peel the fruit to acquire the NIR spectrum.

6.- Lines 163-164: The equipment allows to modify these parameters easily?

Author Response

Dear Editor and Reviewers,

We sincerely appreciate your consideration and the reviewers’ valuable comments concerning our manuscript entitled "Dynamic nondestructive detection models of apple quality in critical harvest period based on near infrared spectroscopy and intelligent algorithms (Foods-3002575)". In general, we found the reviewer’s comments to be very constructive and insightful. The helpful criticism has enabled us to present a better and more informative manuscript.

We have presented the responses directly after each comment raised by the reviewer and the revised portions in the manuscript are marked in red. Our responses appear in italics below.

Responses to Reviewer #1:

General comment:

The article entitled "Dynamic nondestructive detection models of apple quality in critical harvest period based on near infrared spectroscopy and intelligent algorithms " presents a very interesting study and proposes a green and portable method to determine the quality of apples in situ. However, the description of the different sections of the article is not clear. The authors write in a very confusing way, making it very difficult to read and understand many of the sections, starting with the abstract and ending with the discussion. The authors should write it more clearly so that it can be fully understood.

Response: Thank you for the careful reading of the manuscript and the constructive comments. As you suggested, in the revised MS, we have made many modifications and improved the logic and readability of the language. We hope that the responses provided for each comment below have addressed the Reviewer’s concerns and significantly improved the revised manuscript.

Comment 1:

In line 20 the acronym "SSC" appears for the first time and there is no indication of what it means. I assume it means "Soluble Solids Content". It is an important parameter in the development of the article and at no time do the authors indicate what it means.

Response: Thank you for your careful comments. We have explained “SSC”, SSC have been changed to soluble solids content (SSC) (line 20-21).

Comment 2:

Line 28 indicates that the colour is measured by PLS-LDA. As far as I know, PLS-LDA is not an analytical technique for measuring colour.

Response: Thank you for your questions. PLS-LDA is essentially a feature-variable-based classification method that can be applied to near-infrared spectra. Zheng used visible/near-infrared spectroscopy for online detection of granulation of sweet orange pulp using the standard normal transform-mean centering-CARS-PLSDA model with the best results in identifying granulated sweet oranges, with an identification accuracy of 94 %. In this paper, the L*a*b* values of apple peel color were used to discriminate between low and high ripeness to classify the spectra, and the classified spectra were pre-processed by SG-SNV for PLS modeling. The PLS model predictions of classified low and high ripeness apples were slightly improved compared to the previous ones.

Reference:

Zheng, Y.; Tian, S.; Xie, L. Improving the identification accuracy of sugar orange suffering from granulation through diameter correction and stepwise variable selection. Comput. Electron. Agric. 2023, 200, 112313.

Comment 3:

Lines 95-97: In the study, are samples from localized production and samples purchased in supermarkets used?

Response: Thank you very much for your inquiry. The apples used in this experiment were produced in an orchard at 80°16' east longitude and 41°18' north latitude in Aksu region, Xinjiang. The apple samples were all collected directly from the orchard and no locally produced or supermarket purchased samples were used.

Comment 4:

Lines 115-116: Indicate characteristics of the black and white standard. Indicate why it is necessary to calibrate? I think the authors are referring to adjusting the equipment.

Response: Thank you very much for your comment. The spectrophotometer was first calibrated with a standard white calibration plate CM-A145. The details of the instrument have been supplemented in the revised manuscript (line 124-125).

Comment 5:

Section 2.5: It is not clear if it is necessary to peel the fruit to acquire the NIR spectrum.

Response: Thank you very much for carefully reviewing. The aim of our study is to utilize the Apple Internal Quality Handheld Detector for nondestructive assessment of internal apple quality using NIR spectroscopy. During spectral acquisition, we securely fix the apples on rubber washers and initiate the spectral acquisition software. The LED light source directly irradiates the surface of the apples. Once the spectral data appears in the software interface, we complete the spectral acquisition process. Our operations are conducted on the surface of the apples, eliminating the need for peeling. The details of the instrument have been supplemented in the revised manuscript (line 180-181).

Comment 6:

Lines 163-164: The equipment allows to modify these parameters easily?

Response: Thank you very much for your inquiry. As described in the materials section, the parameters such as the number of scans, exposure time, sampling frequency, and PWM value were optimized before the experiment and set to consistent values (average number of scans = 5, exposure time = 75 ms, sampling frequency = 10 Hz, PWM value = 60). However, the equipment allows these parameters to be modified easily during measurements to adapt to varying environmental conditions. This flexibility ensures the robustness and accuracy of the measurements.

 Finally, I would like to thank the editors and reviewers again for their valuable comments and suggestions.

Reviewer 2 Report

Comments and Suggestions for Authors

The manuscript entitled “Dynamic nondestructive detection models of apple quality in critical harvest period based on near infrared spectroscopy and intelligent algorithms”, authored by Zhiming Guo, Xuan Chen, Yiyin Zhang, Chanjun Sun, Heera Jayan, Usman Majeed, Nicholas J. Watson and Xiaobo Zou, presents a practical utilization of chemometric methods in fruit (apples) quality control. The manuscript is clearly written, well-structured and contains valuable results. The paper should be accepted for publication, however some issues should be addressed.

1.      TITLE: The title is well constructed and reflects the main purpose of the study.

2.      ABSTRACT: Well-written, points out the main purpose of the study, main results and conclusions.

3.      INTRODUCTION: Nicely written. The relevant literature has been cited. All the necessary background has been covered. Line 74: “According to previous studies…” (what studies authors are referring to?) Line 77: “Previous studies…” (what studies authors are referring to?).

The objectives of the study are clearly listed in the last paragraph.

4.      MATERIAL AND METHODS: The authors listed and clearly presented all the methods and materials used in the study. Line 101: “14 days…” (the sentence should not start with a number). Line 110: The first sentence sounds unfinished.

It is not quite clear how the PLS model was validated (except the calculation of statistical parameters). Was the internal or external validation procedure applied? Was the external test set formed? If the external validation was not applied, I suggest to use it to confirm the real predictive power pf the PLS model.

Lines 229-230: It should be: Rc – correlation coefficient of calibration; RMSEC – root mean square error of calibration; Rp – correlation coefficient of prediction

5.      RESULTS: Figures 4 and 5 should be improved in terms of resolution. Generally, the results are well presented.

6.      DISUCSSION: Well written, but validation of the PLS model should be better elaborated.

7.      CONCLUSIONS: I suggest adding the recommendations for further investigation.

Comments on the Quality of English Language

I have noticed several minor mistakes that should be corrected.

Author Response

Dear Editor and Reviewers,

We sincerely appreciate your consideration and the reviewers’ valuable comments concerning our manuscript entitled "Dynamic nondestructive detection models of apple quality in critical harvest period based on near infrared spectroscopy and intelligent algorithms (Foods-3002575)". In general, we found the reviewer’s comments to be very constructive and insightful. The helpful criticism has enabled us to present a better and more informative manuscript.

We have presented the responses directly after each comment raised by the reviewer and the revised portions in the manuscript are marked in red. Our responses appear in italics below.

Responses to Reviewer #2:

General comment:

The manuscript entitled “Dynamic nondestructive detection models of apple quality in critical harvest period based on near infrared spectroscopy and intelligent algorithms”, authored by Zhiming Guo, Xuan Chen, Yiyin Zhang, Chanjun Sun, Heera Jayan, Usman Majeed, Nicholas J. Watson and Xiaobo Zou, presents a practical utilization of chemometric methods in fruit (apples) quality control. The manuscript is clearly written, well-structured and contains valuable results. The paper should be accepted for publication, however some issues should be addressed.

Response: Thanks to the reviewer for your careful reading and affirmation of our work, we hope that the responses provided in each of the following comments will address the reviewer’s concerns and significantly improve the revised manuscript at the suggestion of the reviewer.

Comment 1:

INTRODUCTION: Line 74: “According to previous studies…” (what studies authors are referring to?) Line 77: “Previous studies…” (what studies authors are referring to?).

Response: Thanks for your helpful comments.

(1) Fruit maturity indicator are soluble solids, titratable acid, and color. Shao et al[20] investigated the relationship between soluble solids and near-infrared hyperspectral data of medium-ripe and fully ripened winter jujubes. Yu[21] et al. constructed a quantitative maturity model for Kurrer balsam pears by choosing hardness, soluble solids content, and titratable acid  to quantify the maturity of pears. Pourdarbani et al. [22] classified “Fuji” apples into four stages of maturity based on color and spectral data (line 75-81).

(2) Zhang et al. [23] collected spectral data of apples to evaluate the maturity levels. On the other hand, DeLong et al. [24] used a handheld chlorophyll meter for chlorophyll content data in apples and the developed model had succesfully asses harvest maturity of 'Minneiska' apples (line 83-86).

Reference:

Wang, F.; Zhao, C.; Yang, H.; Jiang, H.; Li, L.; Yang, G. Non-destructive and in-site estimation of apple quality and maturity by hyperspectral imaging. Comput. Electron. Agric. 2022, 195, 106843.

Shao, Y.; Ji, S.; Xuan, G.; Wang, K.; Xu, L.Q; Shao, J. Soluble solids content monitoring and shelf life analysis of winter jujube at different maturity stages by Vis-NIR hyperspectral imaging. Postharvest Biol. Technol. 2024, 210, 112773.

Yu, S.; Tang, Y.; Lan, H.; Li, X.; Zhang, H.; Zeng, Y.; Niu, H.; Jin, X.; Liu, Y. Construction method of quantitative evaluation model for the maturity of Korla fragrant pear. Int J. Agr Biol Eng. 2022, 15, 243-250.

Pourdarbani, R.; Sabzi, R.; Kalantari, D.; Karimzadeh, R.; Ilbeygi, E.; Arribas, J.I. Automatic non-destructive video estimation of maturation levels in Fuji apple (Malus Malus pumila) fruit in orchard based on colour (Vis) and spectral (NIR) data. Biosys. Eng. 2020, 195, 136-151.

Zhang, B.; Zhang, M.; Shen, M.; Li, H.; Zhang, Z.; Zhang, H.; Zhou, Z.; Ren, X.; Ding, Y.; Xing, L.; Zhao J. Quality monitoring method for apples of different maturity under long-term cold storage. Infrared Phys.Technol. 2021, 112, 103580.

DeLong, J.; Harrison, P.; Harkness, L. An optimal harvest maturity model for ‘Minneiska’ apple fruit based on the delta-absorbance meter. J. Horticult. Sci. Biotechnol. 2020, 95, 637-644.

Comment 2:

Line 101: “14 days…” (the sentence should not start with a number). Line 110: The first sentence sounds unfinished.

Response: Thank you for your helpful suggestion.

(1) Then, 14 days later, when the harvest standard was reached as identified by a professional harvester, samples that were relatively the same size as those of the low-maturity apples and had no scars on their surfaces were detected in-situ on the tree. (line 109-112).

(2) The color of the sample was measured with a spectrophotometer (CM-2300d, Konica Minolta, Osaka, Japan). (line 119-120).

Comment 3:

It is not quite clear how the PLS model was validated (except the calculation of statistical parameters). Was the internal or external validation procedure applied? Was the external test set formed? If the external validation was not applied, I suggest to use it to confirm the real predictive power pf the PLS model.

Response: Thank you very much for your inquiry. Apple samples were randomly divided into calibration and prediction datasets, with a ratio of 3:2, as shown in Table 3. We use a prediction set that is completely independent of the calibration set (data that does not overlap with the calibration dataset) to evaluate the performance of the model. This helps to assess the model's ability to generalize over completely new data. The details of the instrument have been supplemented in the revised manuscript (line 279-281).

Comment 4:

Lines 229-230: It should be: Rc – correlation coefficient of calibration; RMSEC – root mean square error of calibration; Rp – correlation coefficient of prediction

Response: Thank you for your suggestion. Correlation coefficient of calibration (Rc), root mean square error of calibration (RMSEC), correlation coefficient of prediction (Rp), root mean square error of prediction (RMSEP), and residual prediction deviation (RPD) were used to compare and evaluate the performance of the model [37]. The details of the instrument have been supplemented in the revised manuscript (line 241-243).

Comment 5:

RESULTS: Figures 4 and 5 should be improved in terms of resolution. Generally, the results are well presented.

Response: Thank you very much for carefully reviewing. We have improved the resolution of Figure 4 and Figure 5. The details of the instrument have been supplemented in the revised manuscript (Figure 4 and Figure 5).

Comment 6:

DISUCSSION: Well written, but validation of the PLS model should be better elaborated.

Response: Thank you for your careful comments. We have added some details and corrected some errors. The details are as follows:

(1) The apple samples were randomly divided into mutually independent calibration and prediction sets. Among the five pretreatment methods (SG, SNV, MSC, Center and 1stD) SG and SNV performed better. (line 420-422).

(2) Most of the characteristic wavelength variables selected by the CARS algorithm were related to the internal chemical properties of the apples to improve the performance of the model, but potentially valid information was also removed (line 428-431).

(3) The ACO algorithm adaptively adjusted the search behavior through pheromone updating and showed strong adaptability to environmental changes, so it can always find a better solution. Meanwhile, the best results were achieved by using diffuse reflectance spectroscopy combined with the ACO algorithm to predict the SSC of apple at high maturity, with Rp, RMSEP and RPD was 0.88, 0.5678°Brix and 2.466, respectively (line 436-440).

Comment 7:

CONCLUSIONS: I suggest adding the recommendations for further investigation.

Response: Thank you for your objective comments and good suggestion. We added recommendations for further future investigations to our conclusions. This can be expanded to other varieties of apples in the future to verify the generalization and applicability of the model (line 462-463).

Comment 8:

Comments on the Quality of English Language. I have noticed several minor mistakes that should be corrected.

Response: Thank you for your constructive suggestions. We invited two native English-speaking postdoctors to help correct language issues in the MS by revising the entire text over.

 Finally, I would like to thank the editors and reviewers again for their valuable comments and suggestions.

Round 2

Reviewer 1 Report

Comments and Suggestions for Authors

Following the corrections and clarifications that the authors have made to the text, I consider the article publishable in its present form. 

Reviewer 2 Report

Comments and Suggestions for Authors

The authors have corrected the manuscript according to the suggestions.